# Tussilagone Reduces Tumorigenesis by Diminishing Inflammation in Experimental Colitis-Associated Colon Cancer

**DOI:** 10.3390/biomedicines8040086

**Published:** 2020-04-11

**Authors:** Sang-Hyeon Nam, Jin-Kyung Kim

**Affiliations:** Department of Biomedical Science, Daegu Catholic University, Gyeongsan-Si 38430, Korea; rlawlsrud0818@gmail.com

**Keywords:** tussilagone, colitis-associated colon cancer, NF-κB, Nrf2, apoptosis

## Abstract

Background: Tussilagone, a major component of *Tussilago farfara* L., has anti-angiogenic and anti-inflammatory effects. However, the therapeutic and preventive activity of tussilagone in colitis-associated colon carcinogenesis is unknown. Methods: We intended to investigate the therapeutic effects and the potential mechanism of action underlying the pharmacological activity of tussilagone on colitis-associated colon cancer induced in mice using azoxymethane (AOM)/dextran sulfate sodium (DSS). We injected BALB/c mice with AOM and administered 2% DSS in drinking water. The mice were given tussilagone (2.5 and 5 mg/kg body weight) and colon tissues was collected at 72 days. We used Western blotting, immunohistochemistry and real-time RT-PCR analyses to examine the tumorigenesis and inflammatory status of the colon. Results: Tussilagone administration significantly reduced the formation of colonic tumors. In addition, tussilagone treatment markedly reduced the inflammatory mediators and increased heme oxygease-1 in protein and mRNA levels in colon tissues. Meanwhile, nuclear NF-κB-positive cells were elevated and nuclear Nrf2-positive cells were demised by tussilagone treatment in colon tissues. Tussilagone also reduced cell proliferation, induced apoptosis and decreased the β-catenin expression. Conclusions: Tussilagone administration decreases the inflammation and proliferation induced by AOM/DSS and induced apoptosis in colon tissue. Overall, this study indicates the potential value of tussilagone in suppressing colon tumorigenesis.

## 1. Introduction

Colon cancer is the third most prevalent malignancy worldwide and is the fourth most common cause of cancer-related death [1]. More than one million new cases of colon cancer are reported annually and the incidence rate has been increasing. In 2012, nearly 10% of the total worldwide cancers, 1,361,000 new cases were reported and nearly half died from colon cancer [2]. Common risk factors for colon cancer include genetic background and environmental risk factors, such as diabetes, cholecystectomy, obesity and high-fat diets [3]. Among these factors, long standing inflammatory bowel disease (IBD) is a higher risk for developing colon cancer [4,5,6]. The cumulative risk for developing colon cancer in extensive IBD is a 19-fold increase over that of the general population, which also suggests that chronic intestinal inflammation is a predisposing condition for colon cancer [5]. Indeed, colonic inflammation can simply initiate tumorigenesis and carcinogenesis [4]. For example, numerous immune cells that penetrate the colon enrich the environment for reactive oxygen species and reactive nitrogen species, which impair DNA and accelerate the cancer initiation [4]. Furthermore, inflammatory cells produce large amounts of pro-tumorigenic cytokines, including tumor necrosis factor (TNF)-α and interleukin (IL)-6, which drive tumor progression [7].

Phytochemicals have been drawing increasing attention for cancer prevention because of their chemical diversity, structural complexity, intrinsic biological activity, simple obtainability, affordability, lack of toxic effects and ability to control a variety of signal transduction pathways and cell processes [8,9]. In fact, we reported several laboratory-based studies of the antitumor effects of bioactive phytochemicals [10,11,12,13]

Tussilagone (7R,14R)-14-Acetoxy-7-((2′E)-3′-methylpent-2′-enoyloxy)-oplopanone (Figure 1a), is the major component of *Tussilago farfara* L. (Compositae). *T. farfara* is a perennial herb that prevalently grows in Korea, China, North Africa, Siberia and Europe. The flower buds of *T. farfara* are known as an important folk medicine used in the treatment of coughs and wheezing [14,15]. Until now, there have been few reports about the therapeutic efficacy of tussilagone. Several previous reports described the anti-inflammatory property of tussilagone in both in vivo and in vitro experiments [14,16]. We also reported that tussilagone treatment ameliorates the inflammatory responses in dextran sulphate sodium (DSS)-induced murine colitis [17]. Interestingly, Li et al. showed that tussilagone suppresses colon cancer cell proliferation by promoting the degradation of β-catenin [18]. Based on these data, the question whether tussilagone has any effects on inflammatory related colon cancer arises. Therefore, we evaluated the anti-cancer effect of tussilagone using a mouse model of azoxymethane (AOM)/DSS-induced colitis-associated colon cancer.

## 2. Experimental Section

### 2.1. Reagents and Antibodies

We obtained all reagents, including AOM, from Sigma-Aldrich (St. Louis, MO) unless otherwise indicated. We purchased tussilagone from Avention Chemical (Incheon, Korea), prepared a 50 mmol/L stock solution in dimethyl sulfoxide (Sigma-Aldrich), stored it at −20 °C and then diluted it as needed with a 0.9% normal saline. We purchased DSS (MW 40,000–50,000) from MP Biomedicals (Solon, OH, USA). We did Western blot and immunohistochemistry analyses using the following antibodies (ab): anti-nuclear factor erythroid 2-related factor 2 (Nrf2) rabbit polyclonal ab, anti-heme oxygenase (HO)-1 mouse monoclonal ab, anti-IL-6 goat polyclonal ab, anti-TNF-α mouse monoclonal ab, anti-proliferating cell nuclear antigen (PCNA) mouse monoclonal ab (all Santa Cruz Biotechnology, Santa Cruz, CA), anti-cleaved-Poly (ADP-ribose) polymerase (PARP) rabbit polyclonal ab, anti-NF-κB p65 rabbit polyclonal ab, anti-β-catenin rabbit polyclonal ab, anti-cyclin D1 rabbit polyclonal ab (all Cell Signaling Technology, Danvers, USA), anti-inducible nitric oxide synthase (iNOS) rabbit polyclonal ab (BD Bioscience, Becton Dickinson, USA) and anti-cyclooxygenase (COX)-2 rabbit polyclonal ab (Cayman Chemical, Ann Arbor, MI, USA).

### 2.2. AOM/DSS-Induced Colitis-Associated Colon Carcinogenesis and Design of Drug Treatment

Six-week-old BALB/c mice (weighing 16–18 g) were purchased from Hyochange Science (Daegu, Korea) and maintained at the Laboratory Animal Center of Daegu Catholic University. All animal housing and experimental procedures were reviewed and approved (27 June 2017) by the Institutional Animal Care and Use Committee of Daegu Catholic University (CUC-2017-022).

The procedures of the animal studies were carried out as described in Figure 1b. Briefly, following an acclimation period, mice were randomly and equally divided into four groups, with nine mice per group. They were injected intraperitoneally with 10 mg/kg AOM on day 1 and maintained on a regular diet and water for seven days. After this, 2% DSS was given in the drinking water for five days, followed by regular water for 16 days. We repeated the cycle of DSS and regular water three times. The mice were given tussilagone 2.5 or 5 mg/kg, three times per week, via gastric intubation starting seven days after the AOM injection, until the termination of the experiment. We measured the body weights every week. On day 72, we sacrificed the mice and recovered the colon tissues. The colons were slit open longitudinally along the main axis and washed with phosphate buffered saline (PBS, pH 7.4). The number of tumors was recorded. Subsequently, we fixed a few colon tissues in 4% paraformaldehyde buffer for further histological examination and immunohistochemical analysis.

### 2.3. Histopathology and Immunohistochemistry (IHC)

The distal colon was cut into 5 mm sections and fixed in a 4% paraformaldehyde for 24 h. Paraffin-embedded tissue sections were cut into 5-μm and stained with hematoxylin and eosin (H&E) for microscopic observation. In order to provide a consensus on staining patterns, the sections were assessed by two investigators in a blind manner.

The expressions of NF-κB p65, Nrf2, PCNA and β-catenin in colon tissue sections were assessed by IHC as previously described [13]. Apoptotic cells in colon tissues were detected using an In Situ Apoptosis Detection Kit (R&D Systems, Minneapolis, MN, USA) [13]. Images were taken under a DM500 microscope (Leica Biosystems Richmond Inc., Richmond, IL, USA) and we quantified them by counting the number of positively stained cells in ten randomly selected fields. The number of positively stained cells was expressed as the percent of total cells in the ten fields.

### 2.4. Western Blot Analysis

To isolate whole-cell lysates from colon tissues, we used a PRO-PREP Protein Extraction Solution (iNtRON Biotechnology, Seongnam-Si, Korea) containing a 1x protease and phosphatase inhibitor cocktail (Roche, Indianapolis, IN, USA) and lysed the colon tissues according to the manufacturer’s instructions. In addition, we extracted the nuclear and cytoplasmic proteins from the colon tissues using the NE-PER™ Nuclear and Cytoplasmic Extraction Reagents (Thermo Fisher Scientific Inc., Waltham, MA, USA) according to the manufacturer’s instructions. The protein concentration was quantified using a Bio-Rad protein assay kit (Bio-Rad Laboratories, CA, USA). The lysates (10–30 μg protein) were separated via sodium dodecyl sulfate-polyacrylamide gel electrophoresis and transferred to polyvinylidene fluoride membranes (Roche, Basel, Switzerland). Each membrane was incubated with a specific primary antibody (1:1000) at 4 °C overnight after blocking with 3% skim milk at room temperature for 1 h. After three washes with washing buffer (20 mM Tris-HCl, 500 mM NaCl and 0.1% Tween 20), the membranes were incubated with each suitable secondary antibody at room temperature for 2 h. The ECL Western blot detection reagents (GE Healthcare, Little Chalfont, UK) and Davinch-Chemi CAS-400SM (Davinch-K, Seoul, Korea) were used to visualize the specific protein bands. Band densities were assessed using Total Lab software (Davinch-K).

### 2.5. RT-qPCR Analysis

Total RNA from the colon homogenates was isolated with an RNAeasy mini kit (Qiagen, Hilden, Germany) according to the manufacturer’s instructions. After RNA preparation, cDNA was transcribed using a single reverse transcriptase synthetic step with Superscript reagents (Promega, Madison, WI, USA). The cDNA was used for quantitative PCR analyses using GoTaq^®^ qPCR Master Mix for Dye-Based Detection (Promega, Madison, WI) and LightCycler^®^ Nano (Roche, Basel, Switzerland). The primers used in quantitative PCR were: iNOS, forward 5′-CCT CCT CCA CCC TCC CAA GT-3′ and reverse 5′-CAC CCA AAG TGC TTC AGT CA-3′; COX-2, forward 5′-AAG ACT TGC CAG GCT GAA CT-3′ and reverse 5′-CTT CTG CAG TCC AGG TTC AA-3′; HO-1, forward 5′-CCT CAC TGG CAG GAA ATC ATC-3′ and reverse 5′-CCT CGT GGA GAC GCT TTA CAT A-3′; TNF-α, forward 5′-TGT CTC AGC CTC TTC TCA TT-3′ and reverse 5′-AGA TGA TCT GAG TGT GAG GG-3′; IL-6, forward 5′-CTT CTT GGG ACT GAT G-3′ and reverse 5′-CGC ATT TCC ACG ATT T-3′; GAPDH, forward 5′-TCT TGC TCA GTG TCC TTG C-3′ and reverse 5′-CTT TGT CAA GCT CAT TTC CTG G-3′. We tested the specificity of the reaction by product separation following gel electrophoresis or via a melting curve analysis when the SYBR green was incorporated. We calculated the mean relative expression of the genes and measured the differences using the 2-ΔC(t) method.

### 2.6. Statistical Analysis

All data were expressed as the mean ± SEM. The statistical analyses were performed by a one-way analysis of variance (ANOVA), followed by the Bonferroni post hoc test, using the GraphPad Prism software, version 6.0 (GraphPad, San Diego, CA, USA). *p* < 0.05 denoted statistical significance.

## 3. Results

### 3.1. Tussilagone Treatment Diminishes AOM/DSS-Induced Tumorigenesis in Mice

Because an AOM/DSS mouse model is very similar to the clinicopathological features of human colon cancer and is commonly used to evaluate the efficacies of various drugs, including phytochemicals, we established this model according to the procedures shown in Figure 1b. Since our previous study showed that tussilagone has anti-inflammatory activity at doses ranging from 0.5 to 2.5 mg/kg, we decided to provide a similar dose of tussilagone in this study. As shown in Figure 1c, the mice receiving AOM/DSS lost some body weight after the last DSS cycle. Body weight was gradually restored within the period of normal water. The mean body weight did not differ between the normal control group and tussilagone-treated group (Figure 1c), suggesting the non-toxic effect of tussilagone. As expected, no tumors were found in the normal control-group mice and the AOM/DSS-only group showed the most colonic tumors (26.7 ± 1.7 total tumors per mouse). Tussilagone-administered mice significantly reduced the incidence and size of lesions compared with the AOM/DSS-only group (9.7 ± 1.7 and 7.0 ± 1.8 total tumors per mouse in 2.5 and 5 mg/kg, respectively) as shown in Figure 1d,e. There were significant differences in the tumor size between the AOM/DSS and AOM/DSS/tussilagone-treated groups (Figure 1e). Moreover, the mean tumor load, which was defined as the sum of the diameters of all the tumors in a certain mouse, was significantly reduced in tussilagone-treated mice, as shown in Figure 1f. The colon weight of tussilagone-treated mice seemed less than in the AOM/DSS-only group, although no statistical significance was found (Figure 1g). We also observed that the colon length challenged by AOM/DSS was evidently shorter than in the AOM/DSS/tussilagone group (Figure 1h). Correspondingly, the ratio of colon weight/length was reduced in the tussilagone-treated group compared with the AOM/DSS-only group (Figure 1i). H&E staining also showed that tussilagone administration inhibited the progression of the colon cancer induced by AOM/DSS (Figure 1j). These macroscopic and microscopic data demonstrated that tussilagone apparently suppressed the AOM/DSS-induced colon tumorigenesis.

### 3.2. Tussilagone Treatment Inhibits Inflammatory Responses in Colon Tissues

Since the AOM/DSS-induced colon-cancer model is deeply related to inflammatory response, we next examined whether the suppression of colon tumorigenesis by tussilagone treatment is associated with anti-inflammatory activity via the regulation of pro-inflammatory mediators, including pro-inflammatory cytokines, in colon tissues. Western blot data showed that the protein levels of the inflammatory enzymes, iNOS and COX-2, were effectively reduced by tussilagone administration in the colon (Figure 2a,b). Like the inflammatory enzymes, TNF-α and IL-6 were markedly more elevated in the AOM/DSS-treated mice than in the control group (Figure 2a,b). The elevated protein levels of these cytokines by AOM/DSS treatment were significantly inhibited by tussilagone administration. In contrast, the expression of HO-1 in the colon was significantly induced in tussilagone-treated mice (Figure 2a,b).

Furthermore, the mRNA expression levels of inflammatory mediators and HO-1 were detected in colon tissues. We found that iNOS, COX-2, TNF-α and IL-6 mRNA expressions were significantly up-regulated in the AOM/DSS-treated colons compared to those in the normal control group. Conversely, the mRNA expression of these inflammatory mediators in the colon was significantly lower in the 2.5 and 5 mg/kg tussilagone-administered group. In addition, consistent with protein expression, the mRNA expression of HO-1 in the colons from the AOM/DSS group was dramatically reduced compared with the expression levels observed for the normal control and tussilagone-administered group (Figure 2c).

### 3.3. Tussilagone Treatment Inhibits NF-κB and Activates Nrf2 in Colon Tissues

Since the expression of inflammatory mediators and HO-1 was modulated by tussilagone administration in the AOM/DSS-treated mice, we next examined the activation of NF-κB and Nrf2, transcription factors responsible for the gene regulation of inflammatory mediators and HO-1, respectively, in colon tissues. As shown in Figure 3a,b, the NF-κB levels in nuclei were augmented by the AOM/DSS treatment and were substantially decreased in the 2.5 and 5 mg/kg tussilagone-administered mice. In contrast, the levels of Nrf2 were decreased in the nuclear fraction of colon tissues by the AOM/DSS treatment and restored by tussilagone administration (Figure 3a,b). We also assessed the nuclear expression of NF-κB and Nrf2 by immunohistochemical analysis. Many nuclear NF-κB-positive cells were found in the colons of the AOM/DSS-only group and were substantially decreased by tussilagone administration (Figure 3c,d). These results indicated that the reduced levels of nuclear NF-κB were responsible for the decrease in inflammatory mediators. In addition, the nuclear Nrf2-positive cells of the colon were dramatically reduced in the AOM/DSS-treated mice, but were increased by tussilagone administration dose-dependently, as shown in Figure 3e,f. These results suggest that NF-κB and Nrf2 are responsible for the inhibited expression levels of the inflammatory mediators and the increased levels of HO-1, respectively.

### 3.4. Tussilagone Induces Apoptosis and Inhibits Cell Proliferation in Colon Tissues

As shown in Figure 2, the tussilagone administration inhibited colonic tumorigenesis, so we next examined the role of tussilagone in intestinal epithelial homeostasis, which is controlled by the balance of proliferation and apoptosis in the mucosa. To evaluate the effects of tussilagone on apoptosis, we did a terminal deoxynucleotidyl transferase dUTP nick end labeling (TUNEL) assay to detect apoptotic cells in the colonic epithelium. The results revealed that the number of cells positively stained with TUNEL was significantly increased in the tussilagone-administered groups compared with the AOM/DSS-treated group, indicating that tussilagone promoted apoptosis in the colons of the AOM/DSS-treated mice (Figure 4a,b). Furthermore, the data were correlated with the Western blotting results of cleaved-PARP, a hallmark of apoptosis (Figure 4e,f).

Uncontrolled proliferation is considered a common event during colon carcinogenesis. Accordingly, we examined the expression of PCNA corresponding to the proliferation using immunohistochemical and Western blot analysis. We noticed that there was a marked increase of PCNA expression in the colons of the AOM/DSS-treated mice, but such increases were substantially reduced after tussilagone administration (Figure 4c–f). These results indicate that tussilagone treatment induced apoptosis and reduced the proliferation of epithelial cells in the AOM/DSS-treated colons.

### 3.5. Tussilagone Administration Inhibits β-catenin Activation

A previous study reported that tussilagone blocked the β-catenin/T-cell factor transcriptional activity and down-regulated the β-catenin level in both cytoplasm and nuclei and consequently inhibited the proliferation of colon cancer cells [18]. Therefore, we tested whether tussilagone administration regulated the β-catenin levels in the AOM/DSS-treated mice. As in the in vitro study performed by Li et al. [18], Western blot results demonstrated that administration with tussilagone down-regulated β-catenin in the cytosol and nucleus of colon tissues in the AOM/DSS-treated mice (Figure 5a,b). These results were also confirmed by the immunohistochemical analysis, suggesting that β-catenin signaling is suppressed in response to tussilagone in the AOM/DSS-treated colon tissues (Figure 5c,d). We also tested the effects of tussilagone on the expression of β-catenin target genes, such as cyclin D1. As shown in Figure 5e,f, the expression of cyclin D1 was blocked in response to tussilagone treatment in the colon tissues of the AOM/DSS- treated mice.

## 4. Discussion

The chemopreventive properties of phytochemicals have been attracting more attention over the past decade [8,9], which inspired us to study the chemopreventive potential of various phytochemicals. In this study, we showed that tussilagone, a major component of *T. farfara* L, could suppress colonic tumorigenesis induced by AOM/DSS. We also demonstrated possible mechanisms related to the NF-κB, Nrf2 and β-catenin signaling pathway down-regulated by tussilagone. Thus, our findings suggested that tussilagone might be efficacious against colon cancer.

Until now, reports about tussilagone have been few and far between. For example, Park et al. showed that tussilagone inhibited dendritic cell functions through HO-1 induction [19]. Moreover, the anti-inflammatory properties of tussilagone via HO-1 induction and NF-κB activation were reported in both in vitro and in vivo experiments [14,16,17,18,20]. Li et al. demonstrated the suppressive effect of tussilagone in colon-cancer cells, possibly via controlling the β-catenin degradation [18]. As far as we know, this is the first study to demonstrate the anti-cancer effect of tussilagone in a mouse model of colitis-associated colon cancer.

Inflammation is a factor for the development and progression of various cancers including colon cancer. For example, the natural history of ulcerative colitis patients can be marked by the development of colorectal cancer [1,4,7]. There is also evidence that the risk of colitis-associated colorectal cancer is severely related to the duration and extent of inflammation [1,4,7]. There is no doubt that the regulation of inflammatory responses in the colonic environment is critical for preventing and/or curing colon cancer, because the colitis-associated colon carcinogenesis is possibly encouraged by inflammation. Our study showed that tussilagone administration significantly reduced various inflammatory mediators, such as inflammatory enzymes and pro-inflammatory cytokines, in both protein and mRNA levels. The expression of these inflammatory mediators is regulated by NF-κB, a major transcription factor to regulate tumorigenesis [21,22]. By potentially stimulating the expression of pro-inflammatory genes, NF-κB strongly affects the course of mucosal inflammation and tumorigenesis. In this study, we showed a significant reduction of NF-κB-positive cells in the nuclei of AOM/DSS-treated mice following exposure to tussilagone. These results indicate that the suppressed expression of the pro-inflammatory cytokines and enzymes in tussilagone-administered AOM/DSS-treated colon tissues might be attributed to the inhibition of NF-κB activation.

Nrf2 is a transcription factor, like NF-κB. In mammals, various genes that control the regulation of cellular redox balance, protective antioxidants and phase II detoxification responses are regulated by Nrf2 [23]. The major characteristics of Nrf2 are to some extent mimicked by Nrf2-dependent genes and their proteins, including HO-1, which could improve their cellular antioxidative capacity by producing several antioxidants, such as bilirubin and carbon monoxide, to exert anti-inflammatory effects [24]. Nrf2-deficient mice were susceptible to DSS-induced colitis, because of the reduced levels of antioxidative enzymes, including HO-1, along with the elevated levels of the pro-inflammatory mediators, including IL-6, TNF-α, iNOS and COX-2 [25]. Consistent with our previous study using DSS-induced colitis [17], tussilagone administration in AOM/DSS-treated mice increased the levels of nuclear Nrf2. Furthermore, elevated mRNA and protein levels of HO-1 were found in tussilagone-treated mice. These results suggest that tussilagone exerted its beneficial effects on the AOM/DSS-induced colon cancer via the Nrf2 signaling pathway.

The Wnt pathway plays key roles in determining cell fate during embryonic development, although it also contributes to homeostasis in different tissues in adult life. The ultimate outcome of the Wnt signal is shaped by those genes whose activity is controlled through β-catenin and the T cell factor [26]. When Wnt binds to membranes, heterodimeric receptors inhibit the destruction complexes and subsequently accumulate β-catenin molecules in the cytoplasm. The accumulation of β-catenin leads to its nuclear translocation and complex formation with the transcription factors, T-cell factor/lymphoid enhancer-binding factor, in the nucleus to regulate the expression of the proteins required for cell proliferation, cell cycle regulation, metabolism, migration, lineage commitment and differentiation [26,27,28]. It is well known that the abnormal activation of the Wnt pathway in epithelia is linked to the generation or the progression of carcinomas of the colon, breast, liver, pancreas and others [26]. In particular, the activation of the Wnt pathway is the initial event in a high proportion of colon carcinomas [26]. A previous study by Li et al. indicated that tussilagone is a potential inhibitor of the Wnt/β-catenin pathway through the suppression of β-catenin-dependent transcriptional activity [18]. Tussilagone repressed the Wnt/β-catenin signaling pathway by promoting proteasome-mediated β-catenin degradation. Our study also demonstrated that tussilagone treatment reduced the cytosolic and nuclear levels of β-catenin in the colon tissues and consequently inhibited the colon tumorigenesis of AOM/DSS-treated mice.

Apoptosis is an influential process used as an intrinsic protection mechanism against cancer initiation [29]. Confrontation of apoptosis is the major cause for chemoresistance in most cancers and one of the crucial targets for improving cancer therapy is the regulation of apoptosis-signaling pathways [29,30]. Indeed, the deregulation of apoptosis contributes to the pathogenesis of colon cancer and the resistance to chemotherapeutic drugs and radiotherapy, which act, at least in part, by killing cancer cells [31]. We observed an increased amount of cleaved-PARP and an amplified number of apoptotic cells in the TUNEL staining of colon tissues. These observations show that the anti-cancer effects of tussilagone in colitis-associated colon cancer are partially facilitated by the induction of apoptosis. In addition, since there have been no previous data about whether tussilagone treatment induces apoptosis, this is the first report that demonstrates that tussilagone induces apoptosis.

## 5. Conclusions

To our knowledge, this is the first report to demonstrate the anti-cancer effects of tussilagone in vivo, especially on an AOM/DSS-induced colitis-associated cancer model. Our study demonstrates that tussilagone administration ameliorates colitis-associated tumorigenesis by inhibiting inflammation and inducing apoptosis. Our data might offer evidence for the chemoprevention of colon cancer, since they suggest that tussilagone could be a promising option for colon cancer prevention and treatment.

## Figures and Tables

**Figure 1 biomedicines-08-00086-f001:**
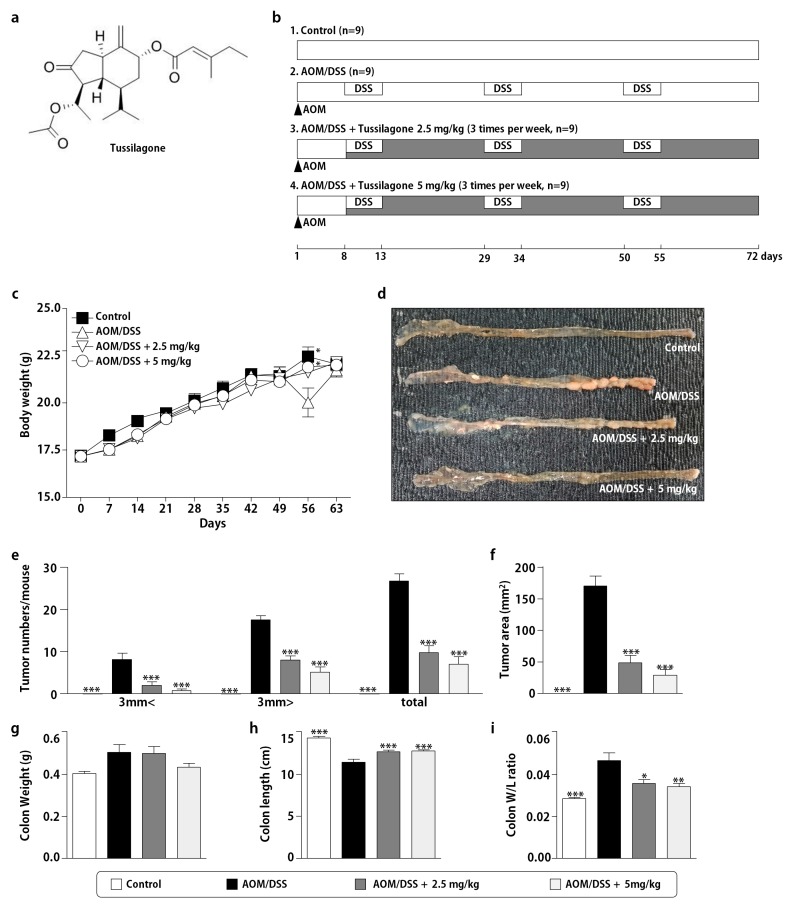
Tussilagone treatment suppressed the AOM/DSS-induced colon carcinogenesis. (**a**) The chemical structure of tussilagone. (**b**) The experimental protocol for the development of a colitis-associated colon carcinogenesis model. (**c**) The weekly recording of the body weights during the experiment and (**d**) the representative macroscopic features of the colons of the mice in each group. (**e**) The number of tumors observed in the colons of mice treated with AOM/DSS and tussilagone. The sizes of tumors were measured and classified as < 3 mm and > 3 mm. (**f**) Total tumor area, (**g**) colon weight, (**h**) colon length and (**i**) the ratio of the colon weight and length per mouse. Results are expressed as the means ± SEM (*n* = 9 per group). * *P* < 0.05, ** *P* < 0.01, *** *P* < 0.001, compared with AOM/DSS-treated mice. (**j**) Histopathology of the colonic mucosa by H&E staining (upper, ×40 magnification; lower, ×100 magnification; bar = 100 µm).

**Figure 2 biomedicines-08-00086-f002:**
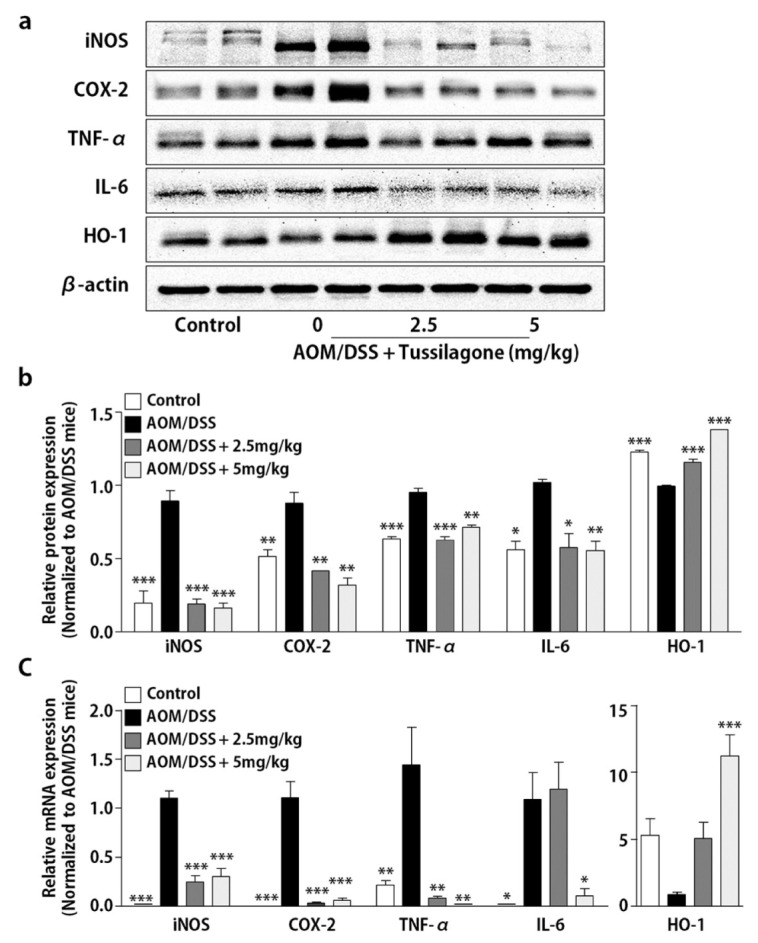
Treatment with tussilagone inhibited AOM/DSS-induced inflammatory mediators in the colon. (**a**) iNOS, COX-2, TNF-α, IL-6 and HO-1 in the colons of mice were detected by Western blot and visualized by chemiluminescence. Representative results are shown and β-actin served as an equal loading control. (**b**) The inflammatory mediator proteins were quantified by densitometric analysis and normalized to β-actin. (**c**) The colonic levels of iNOS, COX-2, TNF-α, IL-6 and HO-1 mRNAs were quantified by real-time RT-PCR and normalized with respect to the mRNA level of the housekeeping gene, GAPDH. Results are expressed as the mean ± SEM. * *P* < 0.05, ** *P* < 0.01, *** *P* < 0.001, compared to the AOM/DSS-treated mice.

**Figure 3 biomedicines-08-00086-f003:**
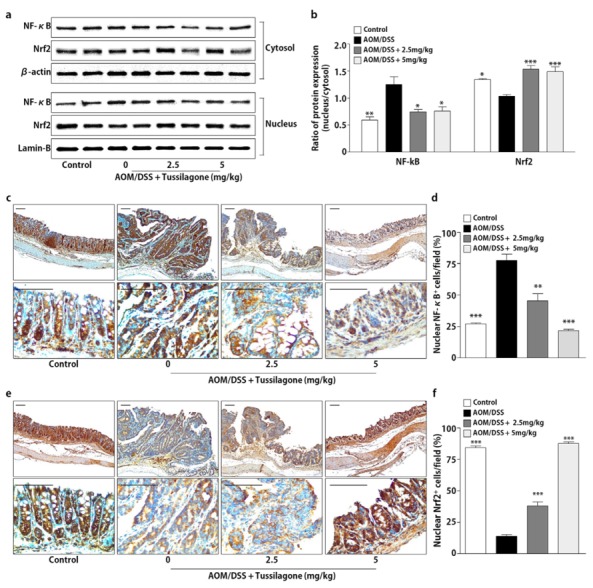
Effect of tussilagone on NF-κB and Nrf2 activities in the colon tissues. (**a**) The protein samples in cytosol or nuclear fraction from the colon tissues were separated by SDS-PAGE. NF-κB and Nrf2 were detected by Western blot and visualized by chemiluminescence. Photos are representative images. (**b**) The quantitative data were normalized by internal control (β-actin for cytosol; lamin-B for nuclear fraction) and further expressed as folds, presented as the comparison with the amount relative to the AOM/DSS-treated mice. (**c**) The immunohistochemical analysis of nuclear NF-κB expression in the colon tissues (upper, ×100 magnification; lower, ×400 magnification; bar = 100 µm). (**d**) The percentage of nuclear NF-κB positive cells in colon tissues was calculated as described in Methods. (**e**) The immunohistochemical analysis of nuclear Nrf2 expression in the colon tissues (upper, ×100 magnification; lower, ×400 magnification; bar = 100 µm). (**f**) The percentage of the nuclear Nrf2 positive cells in colon tissues was calculated as described in Methods. Results are expressed as the mean ± SEM. * *P* < 0.05, ** *P* < 0.01, *** *P* < 0.001, compared to the AOM/DSS-treated mice.

**Figure 4 biomedicines-08-00086-f004:**
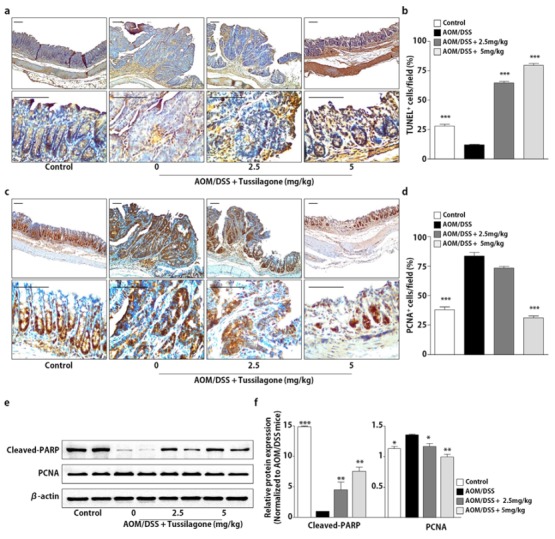
Tussilagone treatment induces apoptosis and inhibits cell proliferation in the colon. (**a**) Apoptosis was measured via terminal deoxynucleotidyl transferase dUTP nick end labeling (TUNEL) staining of the colon sections of mice (upper, ×100 magnification; lower, ×400 magnification; bar = 100 µm). (**b**) The percentage of TUNEL-positive cells in the colon tissues was estimated as described in Methods. (**c**) The cell proliferation was measured via PCNA staining of the colon sections from mice (upper, ×100 magnification; lower, ×400 magnification; bar = 100 µm). (**d**) The percentage of PCNA-positive cells in the colon tissues was calculated as described in Methods. (**e**) The protein samples of the whole-cell lysates from the colon tissues were separated by SDS-PAGE. Cleaved-PARP and PCNA were detected by Western blot and visualized by chemiluminescence. Photos are representative images. (**f**) The quantitative data were normalized by internal control, β-actin and further expressed as folds, presented as the comparison with the amount relative to the AOM/DSS-treated mice. The results are expressed as the mean ± SEM, * *P* < 0.05, ** *P* < 0.01, *** *P* < 0.001, compared with the AOM/DSS-treated mice.

**Figure 5 biomedicines-08-00086-f005:**
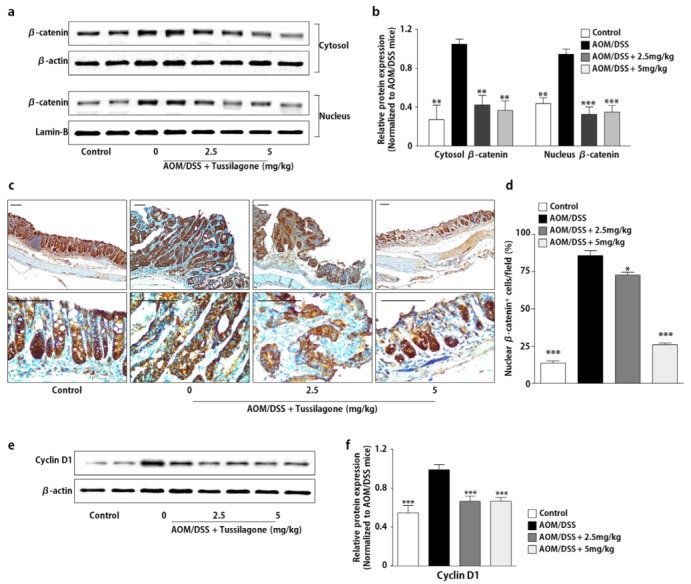
Administration of tussilagone inhibits the β-catenin expression in the colon tissues. (**a**) The protein samples in the cytosol or nuclear fraction from the colon tissues were separated by SDS-PAGE. β-catenin was detected by Western blot and visualized by chemiluminescence. Photos are representative images. (**b**) The quantitative data were normalized by internal control (β-actin for cytosol; lamin-B for nuclear fraction) and further expressed as folds, presented as the comparison with the amount relative to the AOM/DSS-treated mice. (**c**) The immunohistochemical analysis of nuclear β-catenin expression in the colon tissues (upper, ×100 magnification; lower, ×400 magnification; bar = 100 µm). (**d**) The percentage of nuclear β-catenin positive cells in the colon tissues was calculated as described in Methods. (**e**) The protein samples of the whole-cell lysates from the colon tissues were separated by SDS-PAGE. Cyclin D1 was detected by Western blot and visualized by chemiluminescence. Photos are representative images. (**f**) The quantitative data were normalized by internal control, β-actin and further expressed as folds, presented as the comparison with the amount relative to the AOM/DSS-treated mice. The results are expressed as the mean ± SEM, * *P* < 0.05, ** *P* < 0.01, *** *P* < 0.001, compared with the AOM/DSS-treated mice.

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
