# Peer review of "Tussilagone Reduces Tumorigenesis by Diminishing Inflammation in Experimental Colitis-Associated Colon Cancer"

_biomedicines, 2020, doi:10.3390/biomedicines8040086_

Round 1

Reviewer 1 Report

Reviewer's report

Title: Tussilagone Reduces Tumorigenesis by Diminishing Inflammation in Experimental Colitis-Associated Colon Cancer

Date: 28th March 2020

Reviewer's report:

Abstract: In order to demonstrate anti-inflammatory effects of Tussilagone on colitis-associated colon cancer, Nam and Kim induced cancer in mice using azoxymethane (AOM)/dextran sulfate sodium (DSS), and treated the mice with tussilagone (2.5 and 5 mg/kg body weight). Then colonic tissue was collected and analyzed by Western blotting, immunohistochemistry, and real-time RT-PCR. The authors showed that Tussilagone administration reduced significantly the formation of colonic tumors, and markedly reduced inflammatory mediators and increased the main transcription factors for apoptosis. In addition, they observed increased apoptosis and decrease of proliferation in colon tissue induced by AOM/DSS. There is a similar study from the same group (Cheon et al., Chem Biol Interact, 2018), but the results in this study seem to be clear and robust. The manuscript should be accepted after the correction of an editorial error.

Major Compulsory Revisions:

            None

Minor essential Revisions:

  • Line144-155: Editorial error?

Author Response

March 31, 2020

Editor-in-Chief

Biomedicines

Thank you for the kine email

Thank you for your letter dated March 30, 2020 (biomedicines-750245), concerning the status of our manuscript entitled “Tussilagone Reduces Tumorigenesis by Diminishing Inflammation in Experimental Colitis-Associated Colon Cancer." The authors deeply appreciate the reviewer’ constructive comments. We have revised the manuscript to incorporate these comments, as indicated in the following point-by-point responses.

Reviewer's 1:

In order to demonstrate anti-inflammatory effects of Tussilagone on colitis-associated colon cancer, Nam and Kim induced cancer in mice using azoxymethane (AOM)/dextran sulfate sodium (DSS), and treated the mice with tussilagone (2.5 and 5 mg/kg body weight). Then colonic tissue was collected and analyzed by Western blotting, immunohistochemistry, and real-time RT-PCR. The authors showed that Tussilagone administration reduced significantly the formation of colonic tumors, and markedly reduced inflammatory mediators and increased the main transcription factors for apoptosis. In addition, they observed increased apoptosis and decrease of proliferation in colon tissue induced by AOM/DSS. There is a similar study from the same group (Cheon et al., Chem Biol Interact, 2018), but the results in this study seem to be clear and robust. The manuscript should be accepted after the correction of an editorial error.

Thank you for the kind comments. We fixed several editorial error.

Reviewer's 2:

The study “Tussilagone Reduces Tumorigenesis by Diminishing Inflammation in Experimental Colitis-Associated Colon Cancer” by Nam et al. is very interesting piece of work. In this study, the authors have successfully showed that anti tumorigenic effect of tussilagone on colitis-associated colon cancer induced by azoxymethane (AOM)/dextran sulfate sodium (DSS) in mice model. Authors have also shown that reduction of inflammatory mediator by tussilagone plays a pivotal roles in these process. Overall the manuscript is well written and this study reveals the potential to successful clinical application. Therefore this manuscript could be acceptable with an additional clarification.

It is not clear how did the authors select the dose of tussilagone (2.5 and 5 mg/kg body weight) in their experiment. Authors need to clarify the rationale.

Thank you for the kind comments. Since our previous study showed tussilagone has anti-inflammatory activity at doses ranging from 0.5 to 2.5 mg/kg (Ref. 17), we decided to provide a similar dose of tussilagone in this study. We added this information in revised manuscript (Line143-145).

Here is the list that we made a correction

  1. Line 5, 7: We fixed spacing error
  2. Line 9: The scientific name has been changed to italic.
  3. Line 22: Special characters (β) have been changed.
  4. Line23-25: The composition of the sentences has changed slightly.
  5. Line 26: The order of keywords has been changed.
  6. Line 42: Special characters (α) have been changed.
  7. Line 48: The period after “Figure” was removed.
  8. Line 49, 50, 53: The scientific name has been changed to italic.
  9. Line 76~79: We rewrite the whole sentences according to editorial request.
  10. Line 80: The period after “Figure” was removed.
  11. Line 93-102: We rewrite the whole sentences according to editorial request.
  12. Line 109-118: We rewrite the whole sentences according to editorial request.
  13. Line 124: We fixed sentence error.
  14. Line 136-138: We rewrite the whole sentences according to editorial request.
  15. Line 143: The period after “Figure” was removed.
  16. Line 143-145: We explained how the dose of tussilagone was determined upon reviwer’s comments.
  17. Line 145, 148, 152, 153, 154, 156, 157, 159, 160, 161, 162, 181, 182, 185, 193: The period after “Figure” was removed.
  18. Line 196-197: The sentences has changed slightly.
  19. Line 207, 210, 211, 217, 233, 239, 241, 246, 267, 269, 270, 271: The period after “Figure” was removed.
  20. Line 290, 356: The scientific name has been changed to italic.
  21. Line 292, 298, 309, 311, 312, 315, 316: Special characters have been changed.
  22. Line 357: Removed duplicate words (Model).
  23. References: We fixed the format od references.
  24.  

We would like to thank the editor and reviewer for their helpful remarks. We hope that our paper is now ready for publication in Biomedicines.

Sincerely yours,                            

Jin-Kyung Kim, Ph.D.

Professor

Department of Biomedical Science,

Daegu Catholic University,

Gyeongsan-Si Gyeongbuk, Rep.of Korea 38430

Reviewer 2 Report

The study “Tussilagone Reduces Tumorigenesis by Diminishing Inflammation in Experimental Colitis-Associated Colon Cancer” by Nam et al. is very interesting piece of work. In this study, the authors have successfully showed that anti tumorigenic effect of tussilagone on colitis-associated colon cancer induced by azoxymethane (AOM)/dextran sulfate sodium (DSS) in mice model. Authors have also shown that reduction of inflammatory mediator by tussilagone plays a pivotal roles in these process. Overall the manuscript is well written and this study reveals the potential to successful clinical application. Therefore this manuscript could be acceptable with an additional clarification.

It is not clear how did the authors select the dose of tussilagone (2.5 and 5 mg/kg body weight) in their experiment. Authors need to clarify the rationale.

Author Response

(The authors gave the same response as above.)
